# Phase Equilibria and Critical Behavior in Nematogenic MBBA—Isooctane Monotectic-Type Mixtures

**DOI:** 10.3390/ijms24032065

**Published:** 2023-01-20

**Authors:** Jakub Kalabiński, Aleksandra Drozd-Rzoska, Sylwester J. Rzoska

**Affiliations:** Institute of High Pressure Physics Polish Academy of Science, ul. Sokołowska 29/37, 01-142 Warsaw, Poland

**Keywords:** liquid crystals, critical mixtures, critical opalescence, nonlinear dielectric effect, monotectic phase diagram

## Abstract

The transition from the isotropic (I) liquid to the nematic-type (N) uniaxial phase appearing as the consequence of the elongated geometry of elements seems to be a universal phenomenon for many types of suspensions, from solid nano-rods to biological particles based colloids. Rod-like thermotropic nematogenic liquid crystalline (LC) compounds and their mixtures with a molecular solvent (Sol) can be a significant reference for this category, enabling insights into universal features. The report presents studies in 4′-methoxybenzylidene-4-n-butylaniline (MBBA) and isooctane (Sol) mixtures, for which the monotectic-type phase diagram was found. There are two biphasic regions (i) for the low (*TP1*, *isotropic liquid-nematic coexistence*), and (ii) high (*TP2*, *liquid-liquid coexistence*) concentrations of isooctane. For both domains, biphasic coexistence curves’ have been discussed and parameterized. For *TP2* it is related to the order parameter and diameter tests. Notable is the anomalous mean-field type behavior near the critical consolute temperature. Regarding the isotropic liquid phase, critical opalescence has been detected above both biphasic regions. For *TP2* it starts ca. 20 K above the critical consolute temperature. The nature of pretransitional fluctuations in the isotropic liquid phase was tested via nonlinear dielectric effect (*NDE*) measurements. It is classic (mean-field) above *TP1* and non-classic above the *TP2* domain. The long-standing problem regarding the non-critical background effect was solved to reach this result.

## 1. Introduction

Lars Onsager was the first who note and explain the appearance of the transition from the isotropic liquid to the predominantly uniaxial nematic (N) liquid crystalline mesophase as the consequence of the molecular alignment associated with the elongated geometry of elements, only weakly influenced by interactions [1,2]. The spontaneous appearance of the nematic mesophase, with the symmetry-related origin, occurs when the crossover concentration of rod-like elements crossovers a length-to-diameter (*L*/*D*) ratio above some model value. Flory reached a similar conclusion for a lattice model approach, paying more attention to specific attractive interactions [3]. Such general models expectations have been confirmed in numerous experimental systems, from colloids based on suspensions of solid nano and micro rods [4], to solutions of amphiphilic molecules [5], dispersions of high molecular weight molecules [6], mineral colloids [7] or colloidal suspensions of biological particles, for instance, cellulose [8], DNA [9], viruses [10], and further in cellular biomembranes [11].

In this diverse collection of qualitatively different systems, there is one common feature: rod-like uniaxial symmetry of building elements and a universal mechanism of transition from a homogeneous phase, or its analog, to an orientationally ordered nematic-type phase.

Such universality indicates the possibility of studying significant properties in selected experimentally convenient systems and considering the extension of conclusions to the entire category. The ‘natural’ candidate for such a model system with the inherent isotropic-nematic (I-N) transition is rod-like thermotropic nematogenic liquid crystalline compounds (NLC), where fundamental model features are combined with experimental convenience [12,13]. As for beneficial features, one can indicate the link between experimental and theoretical systems, unique phase transitions related to well-defined individual symmetry elements freezing or releasing, or the enormous sensitivity to exogenic and endogenic impacts [12,13,14,15,16,17,18,19]. For exogenic impacts, the most important is the enormous sensitivity to the electric field, which is the driving force for omnipresent innovative applications of thermotropic NLC in displays and photonic devices [20,21].

Notable that some small amount of molecular admixtures/contaminations is the inherent feature of any thermotropic liquid crystalline compound, which can be limited only by deep cleansing [12,13]. The risk of a ‘hidden’ influence of such a factor affecting the properties of the base NLC material is a significant motivation for studies on NLC + molecular solvent (sol) mixtures [12,22,23,24,25,26,27,28,29,30,31,32,33,34,35,36,37,38,39,40,41,42,43,44].

A question arises for its significance in NLC + nanoparticles (NPs) nanocolloids and nanocomposites, where a significant shift of I-N clearing temperature ($T^{C})$ is often reported [45]. It is reported mainly for higher concentrations of NPs, where the additional molecular component serves as a surface agent for nanoparticles, to avoid their sedimentation and aggregation [45,46,47,48,49,50,51,52,53]. Negligible $T^{C}$ shift is observed for small concentrations of NPs, where the additional molecular component is not required [54,55,56,57,58,59,60]. One can expect that for NLC + NPs systems containing the molecular surface agent, some molecules remain ‘free’, and can act as a solvent. It can yield the biphasic domain between the isotropic liquid and nematic phases, finally creating the ‘stretched’ transition between the isotropic liquid and LC mesophases [45,61]. Worth stressing is the importance of LC-based composite systems for new generations of innovative devices [53,62,63,64,65,66,67,68].

This report focuses on the quantitative analysis of relevant features of thermotropic NLC with the addition of a low molecular weight solvent (Sol). First, it is important to recall the fundamental characterizations of the I-N transition in rod-like thermotropic NLC.

Five decades ago, Pierre Gilles de Gennes attempted to describe strong ‘anomalous’ changes of the Cotton-Mouton effect (CME, $\Delta n/\lambda H^{2}$), and Rayleigh’s light scattering ($I_{L}$), on cooling in the isotropic liquid phase of NLC [69,70]. De Gennes formulated the model [69,70,71,72], later referred to as the Landau-de Gennes (LdG) model, combining Landau’s theory showing the dominant role of symmetry changes for continuous phase transitions [73] and the uniaxial’ order parameter appropriate for the I-N transition [72]. LdG model turned out to be extremely important in the *Physics of Liquid Crystals* [12,13,72], *Polymer Physics* [74], and *Soft Matter Physics* [75,76] in subsequent decades. The grand success of LdG model was also associated with its generic universality. All these were important for honoring Pierre Gilles de Gennes with the Nobel Prize in 1991 [75]. In the meantime, it was shown that parallel temperature changes are observed for *CME*, *I_L_* [12,13,77,78] and also the electrooptic Kerr effect (*EKE*), Optical Kerr effect (*OKE*) ($\Delta n/\lambda E^{2}$) [12,78], compressibility ($\chi_{T}$) [79], nonlinear dielectric effect (*NDE*, $\Delta \varepsilon /E^{2}$) [80,81,82]:(1)$$\frac{\Delta n}{\lambda H^{2}},\frac{\Delta n}{\lambda E^{2}},\frac{\Delta \varepsilon }{E^{2}},I_{L},\chi_{T}=\frac{A}{T-T^{*}}$$
where $T>T_{I-N}=T^{*}+\Delta T^{*}$, $T^{*}$ is the extrapolated temperature of a hypothetical continuous phase transition, $\Delta T^{*}\;$ is the I-N transition discontinuity metric; Δn denotes the birefringence induced by the strong electric field *E*, or magnetic field *H*, for *KE* and *CME* respectively; $\Delta \varepsilon =\varepsilon -\varepsilon \left(E\right)$ is the difference of dielectric constants under the weak (measurement) and the additional strong *E* electric fields; λ is the light wavelength and A is the constant amplitude related to the tested magnitude.

By tradition, the phase transition temperature from the isotropic liquid to LC mesophases is also called the clearing temperature ($T^{C}$), to indicate the loss of optical transparency on cooling [12,13]. Following Equation (1), the analysis of reciprocals of listed magnitudes enables simple and reliable estimations of $\left(T^{*},\Delta T^{*},A\right)$ values via the linear regression fit. For other continuous or semi-continuous phase transitions in liquids, such as the gas-liquid critical point, or the critical consolute point in binary mixtures, the above properties show different patterns of pretransitional changes [12,81,82,83,84].

So far, no studies on pretransitional effects for physical magnitudes recalled in Equation (1) have been reported for NLC-Sol mixtures. Studies of such systems are focused mainly on determining phase diagrams (temperature vs. concentration) and their discussion in frames of classic mixtures/solutions models, thermal properties, microscopic insight, and spectroscopic (mainly NMR or IR) studies [37,38,39,40,41,42,43,44]. Pretransitional effects in the isotropic phase were studied only for small solvent concentrations: for the heat capacity ($C_{P}$, [36]) and dielectric constant (ε, [35]). These tests enabled estimations of the heat capacity critical exponent describing pretransitional effects of mentioned properties: α=0.3−0.5, depending on solvent concentration. The impact of solute on $\Delta T^{*}$ was not tested, which can be associated with the relatively large number of fitted (five!) parameters in relations describing $C_{P}\left(T\right)$ and $\varepsilon \left(T\right)$ pretransitional changes.

So far, studies in NLC + Sol mixtures [12,22,23,24,25,26,27,28,29,30,31,32,33,34,35,36,37,38,39,40,41,42,43,44] have shown the decrease of clearing temperature for small solvent concentrations and the emergence of two coexisting phases between the isotropic liquid and the nematic phases. For some NLC + Sol mixtures, a second biphasic domain for high solute concentrations was noted [12,30,34,40,41,42,43,44]. It is bounded by the coexistence curve resembling the binodal observed for binary, non-mesogenic liquid mixtures of limited miscibility. To the best of the authors’ knowledge, there are no attempts for the functional analysis of limited miscibility domains in NLC + Sol mixtures. Notable that following the *Physics of Critical Phenomena* applied for low molecular weight liquids binary mixtures of limited miscibility, the binodal is described by the ‘width’ ($\Delta x\left(T\right)$), related to the order parameter, and the diameter $d\left(T\right)$ [12,85,86]:(2)$$M=\Delta x\left(T\right)=\left|x_{U}\left(T\right)-x_{L}\left(T\right)\right|=B{\left|T_{C}-T\right|}^{\beta }\times \left(1+{\left|T_{C}-T\right|}^{\Delta_{1}}+\ldots \right)$$
(3)$$\;d\left(T\right)=\frac{x_{U}\left(T\right)+x_{L}\left(T\right)}{2}=d_{C}+b{\left|T_{C}-T\right|}^{2\beta }+a{\left|T_{C}-T\right|}^{1-\alpha }+c\left|T_{C}-T\right|$$
where $T<T_{C}$, $T_{C}$ is the critical consolute temperature located at the top of the binodal, M denotes the order parameter, β is the order parameter-related critical exponent, and α denotes the heat capacity-related critical exponent; $\Delta_{1}=0.5$ is the first correction-to-scaling critical exponent necessary away from $T_{C}$.

Values of critical exponents depend only on the space (*D*), and the order parameter (*N*) related dimensionalities. Consequently, pretransitional (pre-critical) effects in microscopically different systems’ are described by isomorphic pretransitional effects as far as they belong to the same universality class (*D, N*). For instance, the same values of critical exponents characterize the surroundings of the gas-liquid critical point in a one-component fluid, the critical consolute point in binary mixtures of limited miscibility, the simple magnetic system with the paramagnetic–Ferromagnetic transition when passing Curie temperatures or their Ising-model parallels [12]. For mentioned system α≈0.11, β≈0.326, and γ≈1.237 for the compressibility (order parameter-related susceptibility) changes. These values recall the basic type of criticality associated with $\left(D=3,N=1\right)$, also called the ‘non-classic’ case. When increasing the range of relevant interaction or the dimensionality, one obtains a so-called ‘classic’ description. One can distinguish two classic patterns. For systems with a single critical point and the dimensionality D≥4: β=1/2 and γ=1, and α=0 for $T>T_{C}$, α=1/2 for $T<T_{C}$: it is also recalled the ‘mean-field’ (MF) behavior. In the case of the tricritical point (TCP) the border dimensionality decreases to D=3, which is associated with the following exponents: β=1/4, γ=1, and α=1/2 both for $T<T_{C}$ and $\;T>T_{C}$ [12]. For I-N transition in thermotropic NLC there is extensive evidence for α=1/2 in the isotropic liquid phase ($T>T_{C}=T^{*}$). For the order parameter exponent, numerous experimental results in NLC indicate β=1/2, including the nematogenic MBBA compound, tested in this report [12,87,88,89]. On the other hand, in octyloxycyanobiphenyl (8OCB) and pentylcyanobiphenyl (5CB) the distortions-sensitive analysis yielded β=1/4 [90,91]. Concluding, the experimental evidence indicates the exponent α>0 in the isotropic phase, but for the exponent β in the nematic phase, both MF and TCP values are reported for different NLC compounds. It may suggest some system-dependent interplay between MF and TCP classic descriptions.

When discussing NLC + Sol phase diagrams, notable is their similarity to the general pattern of monotectic mixtures [92]. Such systems have remained a significant area of significant interest in metallurgy for decades [93,94]. Recently, it has increased thanks to the possibility of obtaining high-entropy alloys (HEAs) with unconventional physicochemical properties [94]. A similar phase diagram was also found in several (non-mesogenic) molecular liquids mixtures. In these studies, the importance of investigating monotectic-type systems for experimentally in a more convenient temperature range than used in metallurgy was the first motivation [95,96,97,98,99]. One can recall succinitrille-pyrene mixtures, for which the characteristic monotectic phase diagram appears in the range of 100–200 °C [97,98]. Monotectic-type mixtures have also been noticed in polymeric mixtures of pharmaceutical importance [100,101,102,103,104]. They were indicated as the base for innovative liquid-liquid extraction technology, tuned by temperature and concentration changes [104].

It may be surprising that the functional/quantitative characterization of two-phase regions, being the hallmark of monotectic systems, is still lacking. Such results could constitute the essential checkpoint for theoretical models and an important prognostic tool in material engineering. As the universal patterns reference, it indicates the significance of limited miscibility studies in monotectic-type mixtures, especially in experimentally convenient systems.

This paper presents the results of studies in an NLC + Sol monotectic type model system: 4′-methoxybenzylidene-4-n-butylaniline (MBBA, rod-like NLC) and isooctane (Sol: the molecular solvent) mixtures, where the limited miscibility occurs in near-room temperatures. The phase diagram and the functional analysis of two-coexisting phase domains for the low- and high- concentrations of isooctane are presented. The latter includes analysis recalling Equations (2) and (3), revealing some unique features. Supplementary nonlinear dielectric effect (*NDE*) studies in the isotropic liquid phase enabled insights into the criticality nature of the above-mentioned biphasic domains. *NDE* refers to changes in dielectric constant $\left(\varepsilon \right)\;$ by the strong electric field, $\Delta \varepsilon /E^{2}=\left(\varepsilon \left(E\right)-\varepsilon \right)/E^{2}$, and it is directly coupled to multimolecular fluctuations [82,84,105,106]. The long-standing problem of the non-critical background *NDE* effect necessary for estimating the critical contribution has been solved. Finally, different types of ‘critical’ opalescence [107,108,109] for low- and high- concentrations of isooctane domains have been detected.

## 2. Results

### 2.1. Phase Diagram

The phase diagram for MBBA—Isooctane mixtures is presented in Figure 1. It reveals two different biphasic domains: *TP1* is related to the coexistence of the nematic and isotropic liquid phases for low concentrations of isooctane, and *TP2* is for high concentrations of isooctane and two isotropic liquid phases coexistence. The inset in Figure 1 focuses on *TP1* domain insight.

Figure 2 shows changes in the temperature width of *TP1* biphasic domain isooctane concentration, also revealing their simple parameterization emerging in the semi-log scale:(4)$$\;ln\left(\Delta T\right)=T_{TP1}^{I}-T_{N}^{TP1}=a+b\times lnx$$
with a=0.2, b=1.0 for x<0.02, and a=1.55, b=6.2 for 0.03<x<0.22; concentrations are given in mole fraction of isooctane.

Notable, that for very small concentration the width of the biphasic domain is very small. It explains the lack of biphasic domain registration in pure materials studies, despite the fact that the presence of residual impurities is a generic feature of LC compounds.

For the *TP2* domain, the temperature quenches from the isotropic liquid phase to the biphasic domain leads to the rapid formation of two coexisting isotropic liquids, resembling the pattern observed for low molecular weight mixtures of limited miscibility [86]. For *TP1* biphasic domain, the process of coexisting phase formation is essentially different. For *TP1* the biphasic domain is associated with the coexistence of LC nematic and isotropic liquid phases. In the given case the formation process requires tens of minutes, enabling the visual registration. Photos showing this process are presented in Figure 3. The numerical filtering enabled insight into the sample’s interior by removing the ‘cloudy/milky’ masking (see Section 4). Such processing reveals the formation of spherical nematic nuclei, whose sedimentation finally leads to the formation of the nematic phase in the lower part of the ampoule. The process of forming coexisting phases is exceptionally long compared to the process of coexistence in low-molecular-weight liquids. Yellow coloring recalls the native MBBA color. The emerging blue color of the upper isotropic phase can be linked to multimolecular heterogeneities/fluctuations of ~400 nm, corresponding to the deep blue wavelength. It is worth recalling that MBBA is 1.6 nm long and isooctane molecule ca. 0.6 nm.

Figure 4 shows the same process of two-coexisting phases formation after the temperature quenches for NLC + Sol samples placed between two parallel glass plates, enabling the polarization-related insight. In this case, there is no gravitational separation of the coexisting phases, as in Figure 3. Note the emergence of nematic ‘seeds/nuclei’, with a visible texture, which number and size increase with the observation time. For such an experimental configuration the assisted opalescence is not detected.

To the best of the authors’ knowledge, there is no discussion regarding the shape of biphasic domains in LC + Sol systems or other monotectic-type mixtures. Such analysis for the MBBA-issooctane system is the key topic of the given report. For the *TP2* domain, it recalls reasoning developed within the *Physics of Critical Phenomena* [12,108,109]. There is no such reference for *TP1* biphasic domain. However, a simple portrayal emerging from the logarithmic plot shown in Figure 2, can indicate that a model description is also possible for *TP1* coexisting phases and such a proposal is also presented.

### 2.2. Liquid-Liquid Binodal

Figure 5 and Figure 6 focus on the binodal coexistence curve for high concentrations of the isooctane domain. They are focused on testing temperature changes of the binodal width related to the order parameter and the binodal diameter—as defined by Equations (2) and (3). These dependencies have been derived within the *Physics of Critical Phenomena* [12,108,109], where universal critical exponents govern the power-type precritical behavior. Their values depend only on space (*D*) and order parameter (*N*) dimensionalities [12]. It allows assembling various systems in their near-critical states into universality classes described solely by (*D*, *N*). For instance, binary mixtures of limited miscibility for low molecular weight liquids belong to $\left(D=3,\;N=1\right)$ universality class, together with the gas-liquid critical point, D=3 Ising model or simple magnetic systems with para⇔ferro transition. For this universality class is associated with the following values of critical exponents: α≈0.110, β≈0.326, the susceptibility (compressibility) related exponent γ≈1.237, and the correlation length exponent ν≈0.625. The same values are expected below and above the critical temperature. For systems with extremely long-range of interaction or dimensionality D>4, obeys the mean-field (MF) approximation for which β=1/2, and γ=1, ν=1/2, and α=0 for $T>T_{C}$, α=1/2  for $T<T_{C}$ [12]. The MF crossover is expected far from the critical point [12,109]. One can also consider the simplified ‘effective’ form of Equation (2), enabling the simple log-log scale analysis [86,110,111]:$$\Delta x\left(T\right)=\left|x_{U}\left(T\right)-x_{L}\left(T\right)\right|\approx B_{eff}{\left|T_{C}-T\right|}^{\beta_{eff}}\;\Rightarrow$$
(5)$$\Rightarrow \;log_{10}\Delta x\left(T\right)=log_{10}B_{eff}+\beta_{eff}log_{10}{\left|T_{C}-T\right|}^{⬚}$$
where $\beta_{eff}>\beta$, and $\beta_{eff}$ is the ‘weighted summa’ of β and $\Delta_{1}$ exponents, depending on the tested temperature range; $\beta_{eff}\to \;\beta$ for $T\to T_{C}$.

Figure 5 presents the log-log plot of the order parameter related $\Delta x\left(T\right)$ changes for MBBA-isooctane binodal vs. the distance from the critical temperature. Such a plot directly recalls the ‘effective’ portrayal via Equation (5).

The results presented reveal the following sequence: $\beta_{eff}\left(close\right)>\beta_{eff}\left(remote\right)$. The typical sequence occurring in low molecular weight liquids is different, namely: $\beta_{eff}\to \beta \approx 0.325$ for $T\to T_{C}$, and on moving away from $T_{C}$ the value of $\beta_{eff}$ increases, typically to $\beta_{eff}\sim 0.36$ due to the impact of the correction-to-scaling terms [12,86]. Well remote from $T_{C}$ the exponent can further increase $\beta_{eff}\to 1/2$, due to the crossover to the mean-field behavior when passing the Ginzburg criterion comparing ranges of key intermolecular interactions and the correlation length [12]. The reversed sequence occurring for MBBA-isooctane binodal is visible in the inset in Figure 5, where the derivative of data from the central part of Figure 5 is tested:(6)$$\beta_{eff}=\frac{dlog_{10}\Delta x\left(T\right)}{dlog_{10}\Delta T_{C}}$$

For discussing the anomalous sequence of $\beta_{eff}$ changes on cooling towards $T_{C}$ in MBBA—Isooctane critical mixture worth recalling is the explanation of the electrooptic Kerr effect (*EKE*) and the nonlinear dielectric effect (*NDE*) behavior on approaching the critical consolute temperature [82,83,84,112], which is based on the emergence of mean-field features near $T_{C}$ due to the uniaxial anisotropy created by the strong electric field. Later, this approach included the I-N transition-related anomaly and, very recently, the strong electric field-induced anomaly on approaching the melting discontinuous phase transition for menthol [113] and thymol [114]. Four decades ago, Beysens et al. [115] tested the shape of nitrobenzene-hexane under shear flow, which also creates the elongation of precritical fluctuations, and observed the ‘anomalous crossover from $\beta_{eff.}\approx 0.36$ remote $T_{C}$ to $\beta_{eff.}\to 1/2$ close to $T_{C}$. Such behavior resembles the pattern presented in Figure 6. The concept of mean-field type behavior associated with the elongation of precritical fluctuations also led to a new model approach for the shear viscosity pretransitional anomaly explanation, on both sides of the critical consolute temperature [116].

Hence, the question arises if the uniaxial anisotropy of MBBA molecules, which can support the similar uniaxial symmetry of pretransitional fluctuations, can lead to the emergence of mean-field behavior hallmarks near $T_{C}$ in the tested MBBA + isooctane mixture?

Figure 6 shows the evolution of the diameter of the MBBA-isooctane binodal coexistence curve. Obtaining unequivocal evidence for the precritical anomaly of the diameter, finally related to Equation (3), constituted the long-standing challenge terminated only in the mid-eighties [12,117,118,119,120]. It can be associated with the relative weakness of the anomaly and the necessity of reliable multi-parameter fitting when using Equation (3) [12,85,86,109]. Earlier, the binodal diameter was considered in the frame of the classical empirical Cailletet–Mather (CM) ‘law’ of rectilinear diameter [12,117,119,121], which can also be derived with mean-field related models:(7)$$d_{CM}=a+b\times T$$

For almost a century, the ‘law’ of rectilinear diameter has been an essential tool for determining the critical concentration in binary mixtures of limited miscibility or critical density for the gas-liquid critical point [99,100,101]. Its meaning comes from the flatness of the upper part of the binodal coexistence curve, i.e., in a relatively broad surrounding of $x_{C}$ or ${}_{C}$ the phase transition temperature can change by no more than 0.01 K. Hence the cancellation of the law of rectilinear diameter yielded a substantial practical problem. The challenge has been solved only recently thanks to a new method based on analyzing the volume of coexisting phases or the fractional meniscus position analysis [86].

Figure 6 explicitly shows the existence of the precritical/pretransitional anomaly of the diameter for the MBBA-isooctane coexistence curve. However, it is so weak and temperature-limited that it could be overlooked without the focused insight in the inset in Figure 7. The solid red curve in Figure 6 is related to Equation (3) with the following parameters: $T_{C}=275.76\;K$, $d_{C}=x_{C}/2=0.335$ m.f., a=−0.21, b=0.16, c=0.60. Due to the weakness of the anomaly, values of critical exponent have to be assumed constant, as for $\left(D=3,\;N=1\right)$ universality class mentioned above.

### 2.3. Critical Opalescence

Critical (pretransitional) opalescence is a famous hallmark of continuous (critical) phase transitions in liquids [12,107,108,109,110,111]. The first observations of this phenomenon were reported by Cagniard de La Tour [107,108] yet in 1822, which is indicated as the onset of *Critical Phenomena Physics* [12,108]. The explanation of this phenomenon by Marian Smoluchowski [122] and Albert Einstein [123] in the early 20th century is recalled amongst the last century’s physics discoveries canon [12,107,108]. The classic evidence for the critical opalescence is for the supercritical region above the liquid-gas phase transition (1) [12,108], but it also appears for the critical consolute temperature in binary liquid mixtures of limited miscibility (2) [86,110,111]. Critical points are located at the top of the binodal coexistence curve in both cases: gas-liquid (1) or liquid-liquid (2) [109]. As the critical point approaches, the size (correlation length) and the lifetime of multimolecular fluctuations diverge [12,108,109]. When their size becomes comparable to the light wavelength, the light is scattered. The transparent liquid becomes turbid, often referred to as ‘milky’ or ‘cloudy’ as the critical opalescence hallmark [12,107,108,109].

Photos presented in Figure 7 shows that the ‘milky’ turbidity also appears in the isotropic phase of MBBA-isooctane mixture in the low-concentrations *TP1* domain for x=0.1 mol f., not expected or evidenced in this domain so far [12]. Notable that for pure MBBA (x=0) such behavior is absent in the isotropic liquid phase. The last ampoule in Figure 7 is for the finally reached biphasic domain.

The second type of phase equilibria appears for high concentrations of the low molecular weight solvent (*TP2*). The biphasic region is bounded by the binodal curve, whose peak can be associated with the critical consolute temperature $\left(T_{C}\right)$. On cooling along the critical isopleth in the isotropic liquid phase, one can finally observe the classic pattern of the critical opalescence [12,107,108,109,110,111] shown in the photos presented in Figure 8. Notable that before becoming ‘milky’, as usually the critical opalescence is indicated [99,100,101], there is a wide temperature range where the color of scattered light changes from deep blue to bluish and milky. Such a pattern was discussed in detail for the nitrobenzene-decane critical mixture in ref. [86]. The unique feature of the results in Figure 8 is the critical onset of the phenomenon, already at $\sim T_{C}+20\;K$.

### 2.4. NDE Insight into Pretransitional Properties in the Isotropic Liquid Phase

In the isotropic liquid phase, the nonlinear dielectric effect (*NDE*) [105,106] tests have been carried for obtaining insight into the character of pretransitional effects, directly coupled to pretransitional fluctuations. Following the model introduced in ref. [82] the fluctuations-related ‘critical’ contribution to *NDE* and *EKE* is described by the following relation [82,106]:(8)$$\frac{\Delta \varepsilon }{E^{2}},\;\frac{\Delta n}{\lambda E^{2}}\propto \Delta M^{2}{}_{V}\chi_{T}\propto {\left(T-T_{C}\right)}^{2\beta }{\left(T-T_{C}\right)}^{-\gamma }={\left(T-T_{C}\right)}^{-\psi }$$

In binary critical mixtures of limited miscibility, a strong electric field induces the uniaxial anisotropy of fluctuations, leading to the mixed-criticality, namely: the order parameter exponent remains non-classic (non-mean-field) and related to (3, 1) universality class, i.e., β=0.325, but for the order parameter related susceptibility (compressibility): γ=1.02. The latter value reflects the classic (mean-field) behavior related to γ=1 with the correction (0.02) appearing near the nonclassic–classic crossover. For *NDE* in critical binary mixtures of limited miscibility, it yields ψ≈0.370. For the isotropic liquid phase of rod-like LC the mean-field type behavior leads to $\langle \Delta M^{2}\rangle {}_{V}=const$ and the exponent ψ=γ=1, in fair agreement with Equation (1) [82,106,112].

Figure 9 presents temperature changes of *NDE* on approaching the I-N phase transition in the isotropic phase of MBBA. The obtained behavior agrees with the first *NDE* test in MBBA, carried out in 1978 [124], but using different measurement concepts and frequencies. The result presented in Figure 9 is related to the LdG model mean-field type behavior described by Equation (1) or Equation (8) within the mean-field approximation. For such behavior, associated with the exponent γ=1, the simple plot of *NDE* reciprocal enables the reliable estimation of the I-N transition discontinuity $\Delta T^{*}=0.8\;K$.

Figure 10 shows that mean-field type *NDE* changes also occur for MBBA-isooctane mixture with* x* = 0.1 m.f. isooctane, i.e., in the mid of low-concentrations (*TP1*) of the isooctane region, where the isotropic liquid and nematic phase are separated by two coexisting phases domain. For this path, the *NDE* pretransitional effect is also related to γ=1, but with the larger value of discontinuity $\Delta T^{*}=3.7\;K$ than in pure MBBA. It can be influenced by the biphasic domain separating the isotropic liquid and nematic phases.

Figure 11 presents the *NDE* increase on cooling towards the critical consolute temperature ($T_{C}=275.76\;K$) along the critical isopleth ($x_{C}$ = 0.67 m.f.) associated with the binodal curve for high concentrations of isooctane domain (*TP2*). The pretransitional effect is of the order $\sim 10^{-19}\;m^{2}V^{-2}$, i.e., close to the smallest registered *NDE* values, for statistical fluctuations of the mean-square of the polarization and density [105,125].

In binary mixtures of limited miscibility, the total registered *NDE* is composed of the critical effect and the non-critical, background ‘molecular’ contribution, namely [126,127]:(9)$$\frac{\Delta \varepsilon }{E^{2}}={\left(\frac{\Delta \varepsilon }{E^{2}}\right)}_{crit.}+{\left(\frac{\Delta \varepsilon }{E^{2}}\right)}_{bckg}=\frac{A}{{\left(T-T_{C}\right)}^{\psi }}+{\left(\frac{\Delta \varepsilon }{E^{2}}\right)}_{bckg}\Rightarrow \approx {\left(\frac{\Delta \varepsilon }{E^{2}}\right)}_{crit.}+a_{b}+b_{b}T$$
where ‘*_crit._*’ is for the critical contribution and ‘*_bckg_*’ is for the molecular background effect.

Generally, the background effect is related to various complex molecular mechanisms [110,126,127]. However, in any case, a linear approximation in a limited range of temperatures is possible, as indicated in Equation (9).

In fact, the experimental determination of the universal *NDE* critical exponent constituted a long-term challenge because of problems in the reliable determination of the background term [82,105,106]. The problem was solved for critical mixtures composed of a dipolar component and a non-dipolar solvent [82,106,125,126,127]. Only the dipolar component introduces a meaningful contribution to the background effect for such mixtures. Hence, *NDE* measurement in a reference mixture of unlimited miscibility (where the critical effect is absent) containing the dipolar component enables the reliable estimation of the noncritical background effect and finally the critical contribution and the exponent ψ. The experimental value obtained in this way ψ≈0.39 [82,126,127] and fairly well agrees with the model predictions mentioned above [82,106]. However, the reference mixture method cannot be applied to critical mixtures where both components significantly contribute to the non-critical background contribution. Unfortunately, it is the case of MBBA-isooctane for the critical isopleth in the *TP2* domain. To solve this grand challenge for *NDE* and also *EKE* studies, one can develop the protocol proposed in ref. [128], and consider the following transformation of experimental data:(10)$$\frac{d\left(\Delta \varepsilon /E^{2}\right)}{dT}=A\psi {\left(T-T_{C}\right)}^{\psi -1}+b_{b}\approx A\psi {\left(T-T_{C}\right)}^{\psi -1}$$
(11)$$\;ln\left[\frac{d\left(\Delta \varepsilon /E^{2}\right)}{dT}\right]=ln\left(A\psi \right)+\left(\psi -1\right)ln\left(T-T_{C}\right)$$

The approximation applied in Equations (10) and (11) is related to the fact that the temperature dependence of the non-critical background term in Equation (9) is very ‘weak’, i.e., the following relation between related coefficients takes place $b_{b}\ll a_{P}$, and consequently the impact of in $b_{P}$ in Equation (10) can be neglected. Consequently, the plot $ln\left[d\Delta \varepsilon /E^{2}\right]$ vs. $ln\left(T-T_{C}\right)$ should yield linear dependence. Such behavior appears for *NDE* pretransitional effect in MBBA-isooctane critical effect, in the insert in Figure 11. The obtained value of the exponent ψ=0.38±0.01, is in superior agreement with the model expectation mentioned above [82,84,106,127]. Notable that in the given case, related to high concentrations of isooctane, the binodal-related phase transition is associated with the isotropic liquid (1)—Isotropic liquid (2) phase equilibria, as for low molecular weight liquids-based mixtures of limited miscibility.

## 3. Discussion

The report presents the results of miscibility studies in nematogenic compound (MBBA) and low molecular weight solvent (isooctane) mixtures. Tests revealed two biphasic domains associated with the isotropic-nematic (I-N) and isotropic liquid–isotropic liquid (*I*_1_–*I*_2_) phase equilibria. For the latter, the limited miscibility is associated with the binodal coexistence curve. The analysis of its shape confirmed the behavior resulting from the *Physics of Critical Phenomena* [12], indicated by Equations (2) and (3). However, some specific features have also been noted. First, it is the anomalous increase of the order parameter exponent in the immediate vicinity of the critical consolute temperature. Second, the diameter of the binodal is relatively close to the Cailletet-Mathias ‘law’ of rectilinear diameter [117,118,119,120,121], showing a very weak precritical anomaly. Both phenomena can be explained as the consequence of the mean-field type behavior emerging close to $T_{C}$, which can be associated with the rod-like structure of MBBA molecules.

As for the biphasic (I+N) domain in low concentrations of the isooctane domain, its width can be effectively portrayed by logarithmic Equation (4), as shown by the inset in Figure 2. However, the question of a possible fundamental reference remains. One can note here that in Equation (4) $\Delta T=T_{I-\left(L,N\right)}-T_{\left(L,N\right)-N}$ plays the role of the ‘coexistence’ width and it gradually decreases (ΔT→0) for x→0 which can be consider as the distance from singularity metric. Hence, ΔT behaves in a way typical for the order parameter [72], and the concentration of isooctane x can describe the distance from ‘pure’ MBBA, i.e., the terminal of the *TP1*-type coexistence. The following parallel of the general order parameter dependence, given by Equation (2), can be considered.
(12)$$M\left(x\right)=\Delta T=T_{I-\left(L,N\right)}-T_{\left(L,N\right)-N}=Ax^{\beta }$$
where *A* is the constant amplitude.

The experimental evidence indicates the mean-field type behavior related to the exponent β=1/2 is associated with the I-N transition in MBBA. Consequently, Equation (12) transforms to:(13)$$\Delta T=Ax^{1/2}\;\Rightarrow \;\Delta T^{2}=A^{2}x$$

The analysis of experimental data based on Equation (13) shown in Figure 12 agrees with the above reasoning. Discrepancy from the linear behavior for x>0.16 can result from the experimental error, but the influence of inherent limitation of concentration ‘x’ as the distance metric is also possible. Distortions very close to ‘pure’ MBBA, for x<0.02 can be associated with a different mechanism for such a small amount of the solute. Notable that I-N transition is a weakly discontinuous transition small value of $\Delta T^{*}\approx 0.8\;K$, as shown from *NDE* studies above. One can state that the ‘critical’ point may be hidden in the experimentally inaccessible space, and in studies pseudospinodal [129] projection is only available.

Photographic observation of the solutions showed the occurrence of critical (transient) opalescence for both discussed biphasic domains. Its existence just above *TP1* domain can support considerations indicating a specific critical-type origin of *TP1* domain.

## 4. Materials and Methods

Studies were carried out using high purity 4′-methoxybenzylidene-4-n-butylaniline (MBBA) with Solid ⇔ 295 K ⇔ Nematic ⇔ 316.2 K ⇔ Isotropic liquid mesomorphism [12,13,87], one of the most ‘classic’ NLC thermotropic compounds, and isooctane as the molecular solvent. Based on structural data from (https://pubchem.ncbi.nlm.nih.gov/, accessed on 12 December 2022), and PyMOL software (https://pymol.org/2/, accessed on 12 December 2022) one can estimate the length of MBBA molecule l=1.75 nm and for isooctane l=0.69 nm. The latter is the non-polar solvent and MBBA is associated with the weak permanent dipole moment approximately perpendicular to the long molecular axis μ=2.6 D [13,130]. Compounds were purchased from Aldrich, with the highest possible purity. MBBA was deeply degassed before sample preparations, reducing eventual parasitic contaminations. The tested mixtures were prepared in a dry box to limit the undesired atmospheric impacts. MBBA—Isooctane mixtures were prepared using the weight and placed in glass ampoules composed of 0.5 ccm spheres, terminated by a tube with 2*r* = 2 mm internal diameter, length 10 cm. The placement of samples into the ampoule was carried out in the isotropic phase using an all-glass syringe with an appropriately long needle. The tops of the tubes were capped using a torch flame. It protected tested samples from external impacts during experiments. Ampoules with samples were placed in *V*~20 L glass vessels with double glass walls. The vessel was connected to a large-volume Julabo thermostat with external circulation. Observations of subsequent phase transitions were made on cooling from the isotropic phase. Observations were repeated a few times to reduce the effect of supercooling, possible for samples with non-critical concentrations related to discontinuous phase transitions. In the homogeneous phase, nonlinear dielectric effect (*NDE*) was carried out [105,106,131]: $\Delta \varepsilon /E^{2}=\left(\varepsilon \left(E\right)-\varepsilon \right)/E^{2}$, where $\varepsilon \left(E\right)$, and ε are for dielectric constants in the presence of the strong electric field *E* and its absence, respectively. *NDE* measurements were carried out using the dual-field method [106,131], in which a capacitor with the tested sample is placed in a resonant circuit associated with the weak measurement electric field, with frequency f≈10 MHz and intensity $E_{m}=10{\;\mathrm{Vcm}}^{-1}$. The strong electric field was applied in as DC pulses lasting Δt=1 ms, and intensities $6{\;\mathrm{kVcm}}^{-1}<E<40\;{\mathrm{kVcm}}^{-1}$. The condition $\Delta \varepsilon \propto E^{2}$ required for *NDE* studies in isotropic liquids was carefully tested. The scheme of the experimental device is given in refs. [106,131]. Studies were associated with *NDE* measurements as small as $10^{-19}m^{2}V^{-2}$, matched of 2–3 digits resolution, which is related to the challenging detection of relative changes of electric capacitance lesser than $\Delta C\left(E\right)/C\sim 10^{-8}$. The *NDE* responses from subsequent strong electric field pulses were cumulated and observed online. It is the order of the lowest value *NDE* ever detected for pretransitional effects [105,106,131]. NDE and coexistence curves were analyzed using the ORIGIN software.

Studies were supplemented by photos showing the opalescence emerging on approaching two-phase regions for the tested NLC + Sol system. For these observations, samples were placed in cylindrical glass tubes, as shown in Figure 13. They were also sealed with a torch flame and put in the mentioned high-volume thermostat- vessel (see refs. [86,111]) to reach the required long-time temperature stability. The thermostatic system is shown in ref. [111].

For the two-phase region associated with low concentrations of isooctane, it was possible to remove the masking impact of turbidity numerically and look “inside” the process, revealing its beautiful and long timescale of formation. The photographs were processed using the GNU Image Manipulation Program (GIMP), an open-source, cross-platform image editing software (https://www.gimp.org/, accessed on 12 December 2022). Processing consisted of rotating and cropping the image and a levels manipulation tool. This tool allows changing the image’s tones through brightness, contrast, and gamma correction. The sole purpose of the operations mentioned above was to highlight the visibility of effects masked by the opalescence. Figure 12 presents the results of such analysis, enabling explicit insight into the formation of the two-phase isotropic-nematic domain.

## 5. Conclusions

The report focuses on the phase diagrams and phase transitions related properties in NLC + Sol, namely for rod-like nematogenic MBBA and low molecular weight solvent isooctane. The monotectic type diagram occurring for near-room temperatures was found. It indicates studies that studies in NLC + Sol system can be a particularly convenient model system for testing universalistic features of monotectic mixtures whose significance extends from metallurgy to polymeric, pharmacy to biotechnology. Notably, it also opens the possibility of high-pressure studies, lacking for monotectic-type systems so far.

Regarding future studies in NLC + Sol mixtures, first one can consider other molecular solvents. The author selected isooctane because existing evidence seems to indicate that for mixtures of LC compounds similar to MBBA and alkanes the domain *TP2* is absent [12], or hidden below the crystallization border. However, it was noted for alkanes isomers [12]. It may indicate the significance of the steric hindrances introduced by the solvent on the discussed phenomenon. On the other hand, available results for rod-like nematogenic pentylcyanobiphenyl (5CB) and water mixtures show the binodal curve appearing with the critical consolute temperature higher than the clearing (I-N) temperature in pure 5CB [40,41,42], indicating the other category of solvents. Regarding LC compounds, promising can be nematogenic and smectogenic n-alkylcyanobiphenyls. Unique phenomena, which can yield features significant for innovative applications, can appear for LC compounds with ‘advanced’ nematic phases, such as the twist-bend nematic phase [132,133,134,135,136] or the ferroelectric nematic phase [137,138,139,140], recently discovered. However, the problem with such tests can be the high costs of novel LC components: such studies as described in the given report, require a few tens ccm of LC compounds:

The report shows that detected *TP1* and *TP2* can be well portrayed within patterns developed by the *Physics of Critical Phenomena*, which have not been reported for any monotectic-type mixture so far. Particularly notable is the link of the *TP1* biphasic domain to critical phenomena, supported by the evidence of critical opalescence in the isotropic liquid phase. The critical opalescence also appears above *TP2* biphasic domain. It is associated with the binodal curve and the critical consolute point, and resembles the pattern observed in low-molecular-weight liquids binary mixtures of limited miscibility. However, it also shows a unique feature: the first hallmark of the opalescence appears even 20 K above the critical consolute temperature.

Associated *NDE* studies have shown the classic (mean-field or TCP) character of pretransitional fluctuations in the isotropic liquid above *TP1* and non-classic above *TP2* domains. This result was achieved due to the solution of the non-critical, molecular background challenge.

The authors would like to stress explicit hallmarks of the supercriticality in the isotropic liquid phase above *TP1* and *TP2* domain, shown by the critical opalescence evidence and *NDE* studies. Supercriticality is a phenomenon of particular importance in modern materials engineering due to the possibility of selective extraction of ingredients or stimulating processes, with intensities that can be precisely tuned and changed by via the distance from the critical point [141]. According to the authors, this report indicates that supercriticality can be a common phenomenon for monotectic mixtures, which can offer innovative supplementation of current applications extending from metallurgy [93,94] to polymer processing in the pharmaceutical industry [100,101,102,103,104].

There is also a question of whether the domains of critical opalescence are not a ‘hidden driving force’ already supporting some processes, e.g., in bio-systems. An example of this is the appearance of a kind of opalescence in biomembranes [142], mainly built from rod-like elements. The results presented in this work can be a significant universalistic reference for all applications-related issues recalled above.

## Figures and Tables

**Figure 1 ijms-24-02065-f001:**
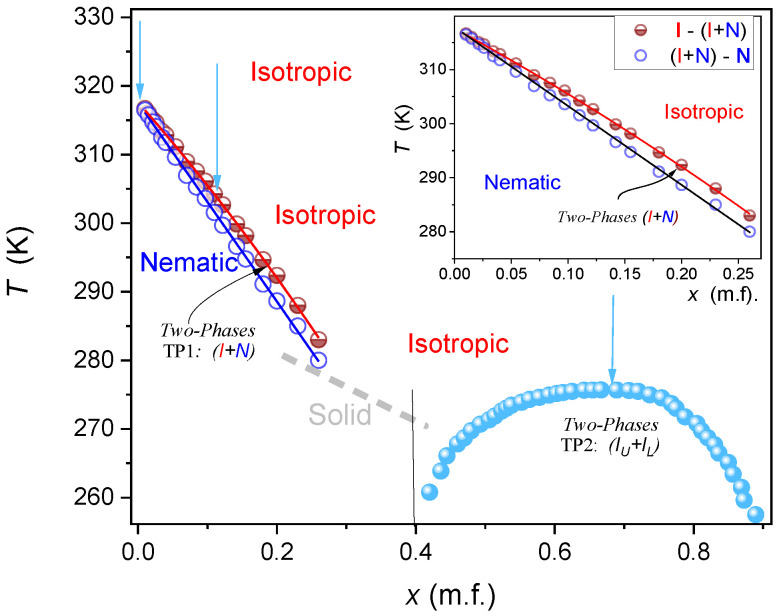
The phase diagram for n-(4-methoxybenzylidene)-4-butylaniline (*MBBA*) nematogenic LC and isooctane mixtures; concentrations are given in mole fractions of isooctane. Vertical arrows show paths of *NDE* studies in the isotropic liquid phase. The inset is focused on *TP1* biphasic domain. Indices ‘_U_’ and ‘_L_’ are for the upper and lower coexisting phases, respectively.

**Figure 2 ijms-24-02065-f002:**
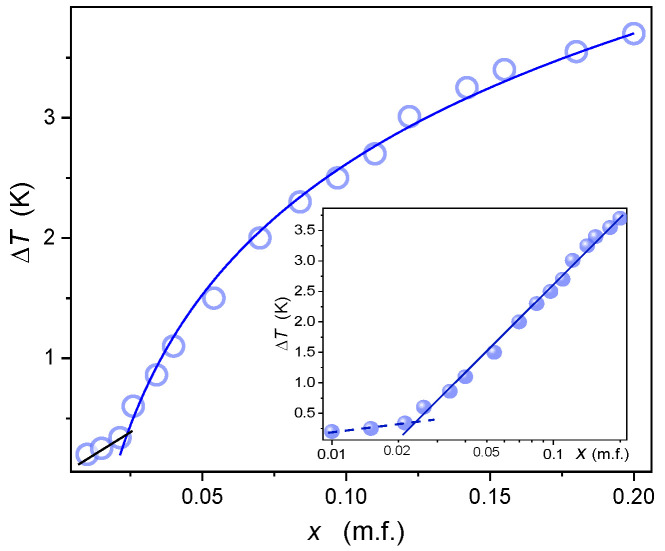
Changes of the temperature width of (I + N) for *TP1* (low concentration of isooctane) biphasic region in MBBA + isooctane mixture; $T=T_{TP1}^{I}-T_{N}^{TP1}$, *x* stands for isooctane concentration in mole fraction. The inset is related to the analysis via the logarithmic Equation (4): the resulting behavior is also shown by the blue curve in the main plot.

**Figure 3 ijms-24-02065-f003:**
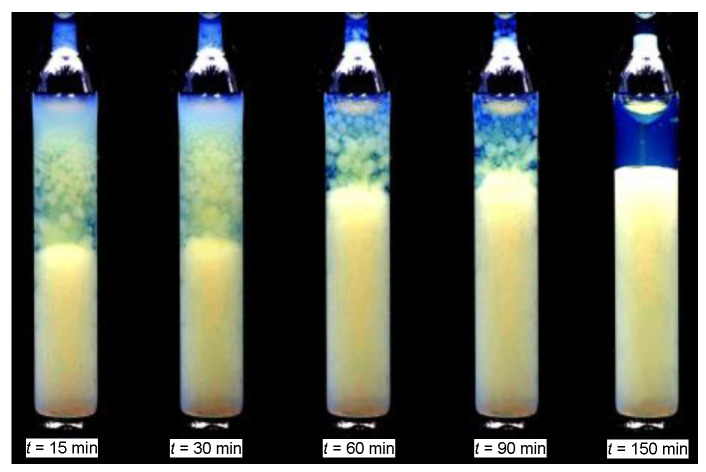
The time-dependent creation of coexisting phases in MBBA—Isooctane mixture two-phase (*2P*) domain: x=0.1 mol f. isooctane, for the temperature $T_{2Ph}^{I}-1\;K$. The photo is after numerical filtering to reach insight into the sample’s interior (see Section 4).

**Figure 4 ijms-24-02065-f004:**
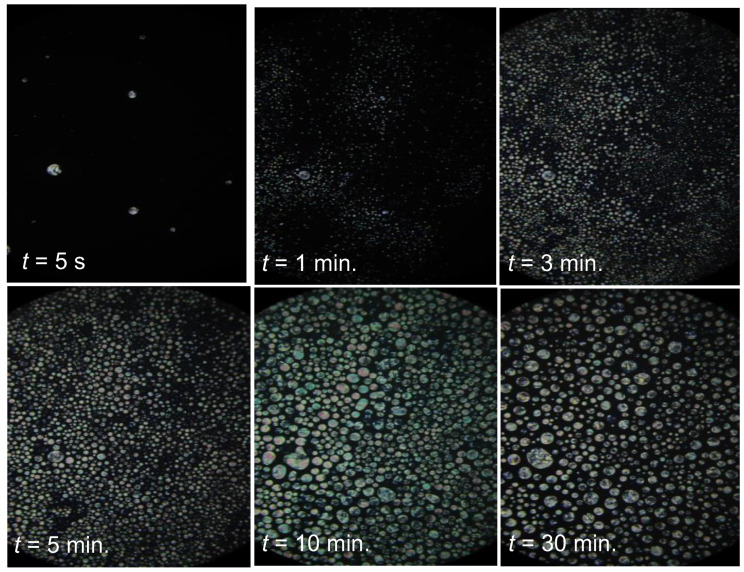
The time-dependent creation of coexisting phases in MBBA—Isooctane mixture two-phase (*2P*) domain: x=0.1 mol f. isooctane, for the temperature $T_{2Ph}^{I}-1\;K$. The photo for samples between two ‘polarizing’ flat glasses, distanced by 0.05 mm. Magnification 10×.

**Figure 5 ijms-24-02065-f005:**
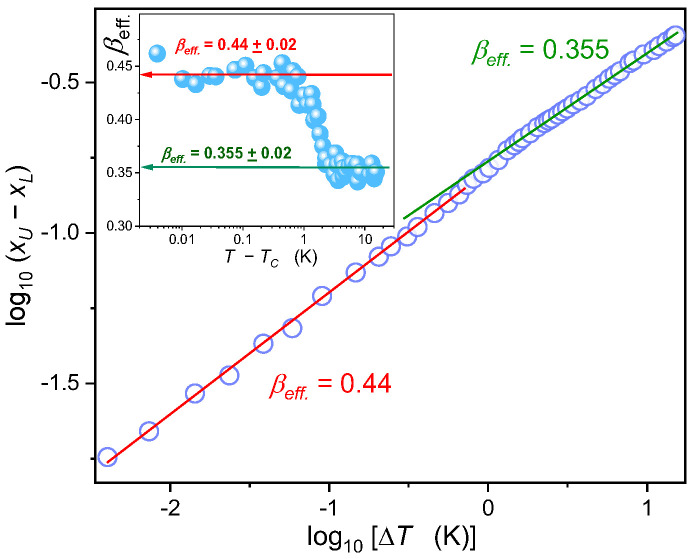
Order parameter related width of the Liquid-Liquid coexistence curve (*TP2* domain) analysis, based on data shown in Figure 1. The applied log-log scale facilitates the analysis via the ‘effective critical’ Equation (5). The inset is for the results of the derivative analysis based on Equation (6). Values of obtained order parameter exponent in subsequent temperature domains are given.

**Figure 6 ijms-24-02065-f006:**
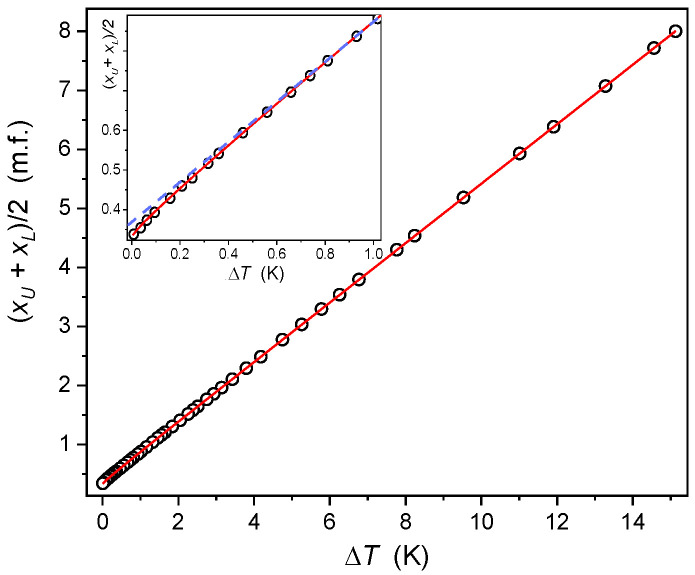
The diameter of the Liquid-Liquid coexistence curve (binodal, *TP2* domain), based on data from Figure 1. The inset focuses on the immediate vicinity of the critical consolute temperature ($T_{C}=276.12\;K$). The solid curve (in red) is related to the portrayal via ‘the critical’ Equation (3). The dashed (blue) line is for extrapolated linear changes, which can portray the behavior for $\left|T_{C}-T\right|>0.6\;K$.

**Figure 7 ijms-24-02065-f007:**
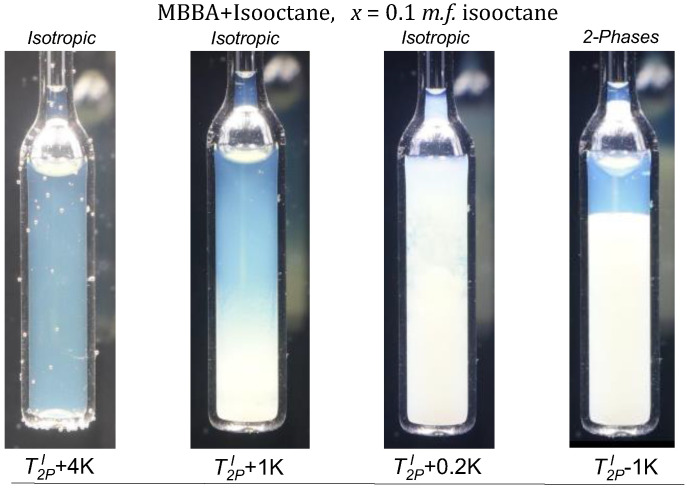
Photos show the pretransitional opalescence in MBBA—Isooctane mixture, *x* = 0.1 m.f. isooctane: in the isotropic (*I*) liquid phase and the two-phase (*TP2*) domains for $T_{2P}^{I}-1\;K$.

**Figure 8 ijms-24-02065-f008:**
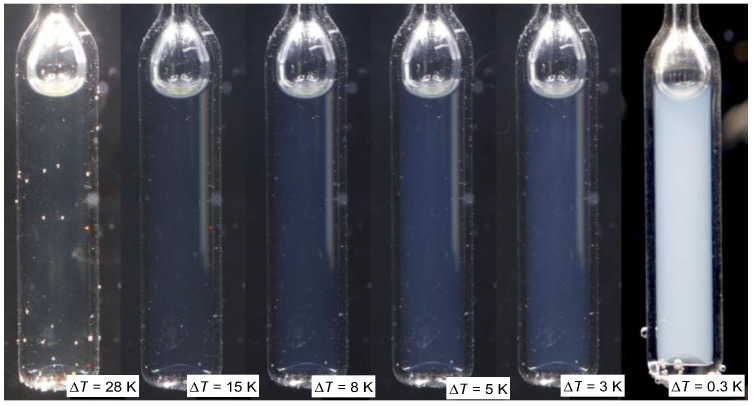
Critical opalescence on cooling towards the critical consolute temperature in MBBA—Isooctane critical mixture, $x_{C}=0.665\;mol.f.$ isooctane; $T=T-T_{C}$ is for the distance from $T_{C}$.

**Figure 9 ijms-24-02065-f009:**
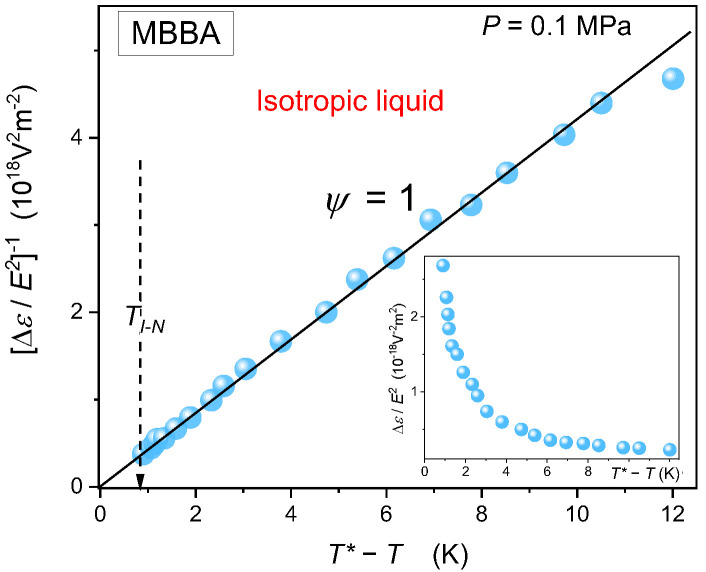
Pretransitional behavior of nonlinear dielectric effect (*NDE*) in the isotropic phase of nematogenic MBBA. The applied scale is related to reciprocals of Equation (1) or Equation (8) with the exponent ψ=1, showing the mean-field type behavior of the pretransitional effect. $T_{I-N}$ and the dashed arrow are for the Isotropic—Nematic weakly discontinuous phase transition (’clearing’ temperature’). The arrow also indicates the value of the discontinuity metric $T^{*}=T_{I-N}-T^{*}=0.8\;K$. The inset shows *NDE* pretransitional effect as detected in the experiment.

**Figure 10 ijms-24-02065-f010:**
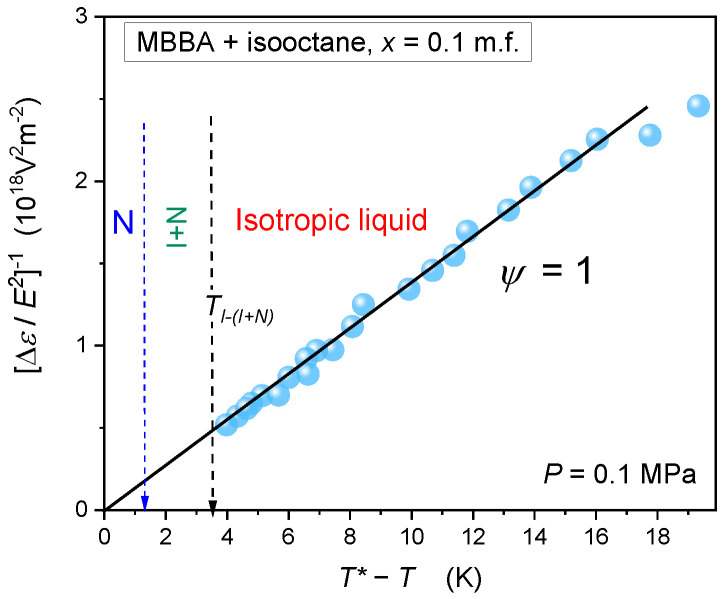
Linear changes of the reciprocal of *NDE* pretransitional effect in MBBA + isooctane mixture (*x* = 0.1 mole fraction), validating the description via Equation (1) or Equation (8) with the exponent *ψ* =1 (mean field approximation). Arrows indicate subsequent phase transitions and the phase transition discontinuity metric: $\Delta T^{*}=T_{I-\left(I+N\right)}-T^{*}$.

**Figure 11 ijms-24-02065-f011:**
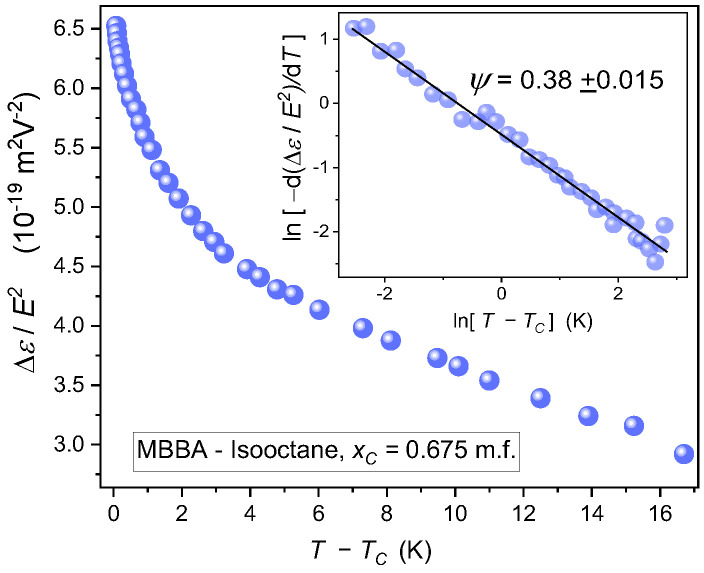
*NDE* pretransitional changes on approaching the critical consolute point in MBBA—Isooctane mixture for the concentration related to the top of the L-L coexistence curve shown in Figure 1. The inset validates the portrayal via the ‘critical’ Equation (8), with the exponent ψ=0.380. It is related to the transformation of data based on Equation (9), which allows the determination of the critical contribution without a biasing impact of the non-critical background effect.

**Figure 12 ijms-24-02065-f012:**
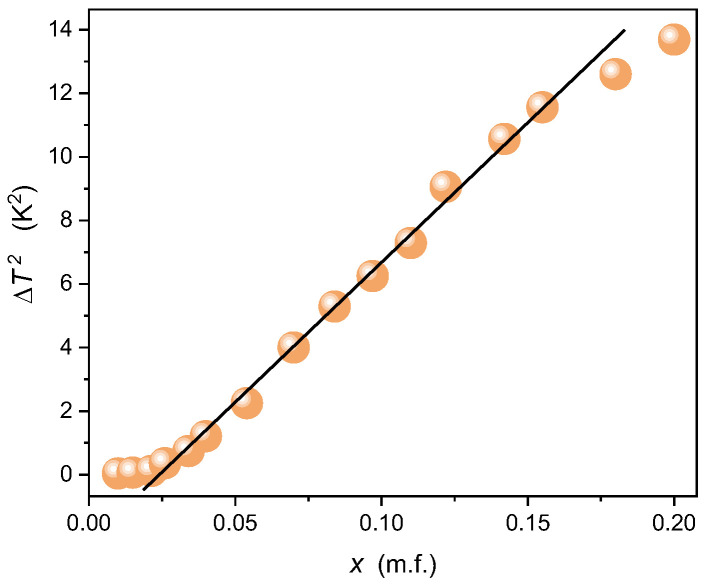
Results of the analysis of *TP1* isotropic-nematic coexistence domain for low concentrations of isooctane (*TP1* domain) validating the portrayal via Equation (10).

**Figure 13 ijms-24-02065-f013:**
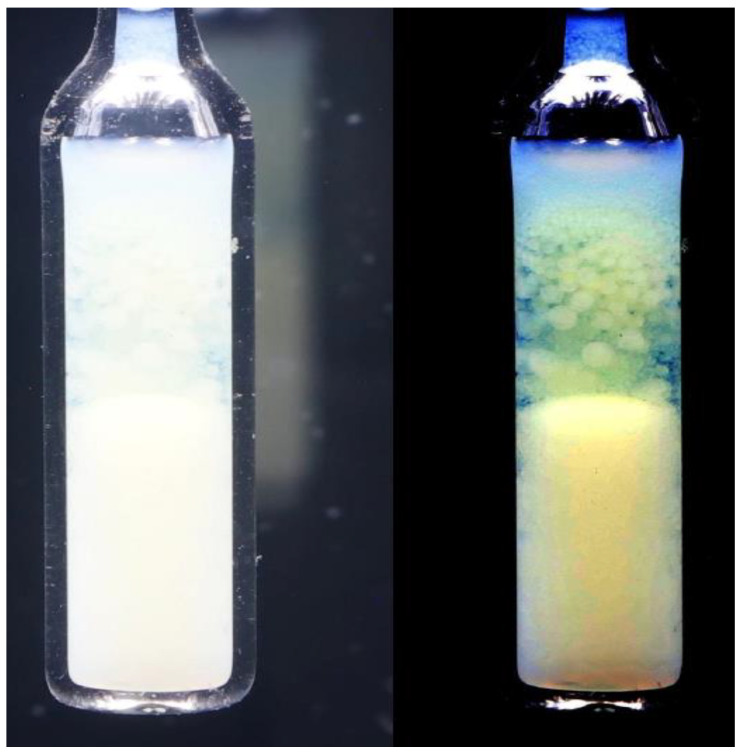
Photos showing two coexisting phases (isotropic liquid and nematic phase) in MBBA + isooctane mixture for *TP1* domain related to low concentrations of isooctane. The photo is for *x* = 0.1 m.f. and the quench temperature $T=T_{2P}^{I}-1\;K$. The left part is for the ‘native’ photo, and the right part is after the numerical filtering, reducing the impact of the opalescence and allowing insight into the sample interior.

## Data Availability

Experimental data are available from the authors on reasonable requests.

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
