# Peer review of "Phase Equilibria and Critical Behavior in Nematogenic MBBA—Isooctane Monotectic-Type Mixtures"

_ijms, 2023, doi:10.3390/ijms24032065_

Round 1

Reviewer 1 Report

Kalabiński et al. investigated the phase equilibria and critical behavior in nematogenic MBBA-
isooctane monotectic-type mixtures to enable the insight into universal features of the Isotropic
(I) to nematic (N) phase transition shown by many suspensions such as solid nanorods and
biological particles-based colloids. They observed two biphasic regions, namely TP1 and TP2
corresponding to the low and high concentration of isooctane in MBBA, respectively. It is
pointed out that TP1 domain is linked with isotropic and nematic coexistence while TP2
domain to the two isotropic liquids coexistence and coupled with binodal curve. The phase
diagram in this system is found to be monotectic type, useful from metallurgy to pharmacy.
The NDE (nonlinear dielectric effect) measurements are found to support the classic mean field
behavior in the isotropic phase of TP1 domain and the non-classic criticality for TP2 domain.
The present study appears to be more fulfilling the fundamental studies along with some
applied work. Therefore, this is a good work and may be published in the prestigious journal,
International Journal of Molecular Sciences (IJMS). However, authors need to refute the
following comments:

1. Authors must check their sentences throughout the manuscript (abstract to conclusion)
as there are many typographical errors and which can not be avoided in any
circumstances.

2. The usual format of writing a manuscript for any scientific journal, we usually follow
the order like Abstract, Introduction, Materials and methods, Results and Discussion,
Conclusion and then references. However, in this manuscript, Results and Discussion
are separated into two separate sections of the manuscript and which I think should not
be the case. Authors have not included the Conclusion in the manuscript, and which is
not acceptable. Moreover, authors should add Materials and Methods section just after
finishing the introduction part unless IJMS prohibits to do so.

3. Under the Materials and Methods section, it is not very clear about the
experimental/computational techniques used for the characterization of investigated
mixtures. Authors must clearly mention the techniques used whether experimental or
computational or both in this section.

4. Authors may cite the paper
https://doi.org/10.1080/10408436.2022.2027226 in page 2
and row 55 of Introduction.

5. Authors may add the following references related to nanoparticles doped liquid crystal
composites in the second paragraph of page 2 of Introduction:

Nanoscale 6, 7743-7756 (2014); Rep. Prog. Phys. 79, 056502 (2016); Progress in

Materials Science 80, 38-76 (2016); Appl. Mater. Today 21, 100840 (2020), J.

Mol. Liq. 297,
1112052 (2020); J. Mol. Liq. 347, 118389 (2022), J. Mol. Liq.
354, 118907 (2022); J. Mol. Liq. 369, 120820 (2023).

6. Authors used the conventional thermotropic liquid crystal compound i.e., MBBA
possessing the regular N phase, as a LC matrix for the preparation of its mixtures with
solvent Isooctane to understand the Isotropic to Nematic phase transition. However,
authors may propose the future work on newly discovered twist-bend nematic and
ferroelectric nematic phases and may cite the following references:

(i) Twist-bend nematic phase: Proc. Nat. Acad. Sci. USA 110, 15931 (2013);
CrystEngComm 17 (14), 27782782 (2015); Journal of American Chemical
Society 138 (16), 5283-5289 (2016); Physical Review E 94 (6), 060701(R)
(2016); Liquid Crystals 44 (1), 191-203 (2017).

(ii) Ferroelectric nematic phase: Proc. Natl. Acad. Sci. U.S.A. 117, 14021 (2020);

Proc. Natl. Acad. Sci. U.S.A. 117, 14629 (2020); Sci. Adv. 7, eabf5047 (2021)
and Proc. Natl. Acad. Sci. U.S.A. 118, e2104092118 (2021).

Author Response

Reviewer #1

  1. Reviewer #1: ‘The usual format of writing a manuscript for any scientific journal, we usually follow the order like Abstract, Introduction, Materials and methods, Results and Discussion,
    Conclusion and then references
    …..’

The answer:  the comment does not agree with IJMS template (available at https://www.mdpi.com/journal/ijms/instructions), in which the following sequence is explicitely given Abstract, 1. Introduction, 2. Results 3. Discussion, 4. Materials and methods, 5. Conclusion,  6. Reference.

Note the comment from the template:

‘5. Conclusions

This section is not mandatory but can be added to the manuscript if the discussion is unusually long or complex.

Hence the paper is prepared following the pattern of IJMS template.

  1. Reviewer #1: ‘…Authors have not included the Conclusion in the manuscript, and which is
    not acceptable. Moreover, authors should add Materials and Methods section just after
    finishing the introduction part unless IJMS prohibits to do so
    ….’

The answer:  Generally, I agree that the Materials and Methods section just after Introduction is a good idea. Notwithstanding, the IJMS template indicates a different pattern (see above), and it is obligatory for the authors. Note also the section ‘Conclusion’ is optional and should be added only if the authors see the necessity (see the instruction for authors) and the template. Hence,  the lack of a Conclusion is acceptable, according to IJMS pattern for the authors.

Notwithstanding, in the revision of the manuscript, the Conclusion section has been added.

  1. Reviewer #1: Under the Materials and Methods section, it is not very clear about the
    experimental/computational techniques used for the characterization of investigated
    Authors must clearly mention the techniques used whether experimental or
    computational or both in this section.

The answer: Note the detailed description in the Methods section, supported by references containing explicit schemes. It is particularly for NDE which is not ‘generally well-known’ method.

  1. Reviewer #1: the addition of a set of additional references has been suggested, namely:
  • https://doi.org/10.1080/10408436.2022.2027226 
  • Nanoscale 6, 7743-7756 (2014); 
  • Prog. Phys. 79, 056502 (2016); 
  • Progress in Materials Science 80, 38-76 (2016); 
  • Mater. Today 21, 100840 (2020), 
  • Mol. Liq. 297, 1112052 (2020); 
  • Mol. Liq. 347, 118389 (2022), 
  • Mol. Liq.354, 118907 (2022); 
  • Mol. Liq. 369, 120820 (2023).

For the above page 2 of Introduction is indicated. In the final part of the report planning of further research, is advised – particularly in respect to ferroelectric LC. Regarding the latter, following references are advised: 

  • Nat. Acad. Sci. USA 110, 15931 (2013);
  • CrystEngComm 17 (14), 2778–2782 (2015); 
  • Journal of American Chemical Society 138 (16), 5283-5289 (2016); 
  • Physical Review E 94 (6), 060701(R) (2016); 
  • Liquid Crystals 44 (1), 191-203 (2017).
  • Natl. Acad. Sci. U.S.A. 117, 14021 (2020);
  • Natl. Acad. Sci. U.S.A. 117, 14629 (2020); 
  • Adv. 7, eabf5047 (2021)
  • Natl. Acad. Sci. U.S.A. 118, e2104092118 (2021).

The answer:  18 additional references suggested by the Reviewer have been included in the paper.

In summary: 

  • Only changes that explicitly followed (all) reviewers' comment has been introduced
  • The paper has been ‘deeply cleaned.’
  • Following comments, the new Figure (current Fig. 4) has been added.
  • Following comments ‘Results’ section has been re-organized into 4 subsection
  • The “Conclusions’ section has been added. It particularly addresses such issues as possible future studies and innovative applications – as suggested by the Reviewer .
  • There are 18 additional references, indicated by Reviewer 1.

Reviewer 2 Report

I think this paper is very good and worth publishing. A small suggestion for your consideration.   

1) There is no core quantitative data in Abstract;   

2) There is no subsection in the second section;   

3) The innovation of the paper needs to be emphasized;   

4) Lack of conclusion.

Author Response

Reviewer # 2

Reviewer # 2: The Reviewer supported the paper, indicating the possible consideration of the following issues:

1) There is no core quantitative data in Abstract;  2) There is no subsection in the second section;   

3) The innovation of the paper needs to be emphasized;   4) Lack of conclusion.

The answer:  The abstract has been supplemented, and subsections have been introduced. Now, the paper also contains the Conclusion section, in which Innovative issues are indicated and stressed. There are also subsections in the second Section.

We would like to stress that Conclusions section is not obligatory, according to the instruction for the authors.

In summary: 

  • Only changes that explicitly followed (all) reviewers' comment has been introduced
  • The paper has been ‘deeply cleaned.’
  • Following comments, the new Figure (current Fig. 4) has been added.
  • Following comments ‘Results’ section has been re-organized into 4 subsection
  • The “Conclusions’ section has been added. It particularly addresses such issues as possible future studies and innovative applications – as suggested by the Reviewer .
  • There are 18 additional references, indicated by Reviewer 1.

Reviewer 3 Report

The phase diagram analysis of the LC composite would be a great interest still because non-trivial applications might be possible by tailoring the LC with guest particles. Here, the authors attempted to study the mixture of nematic LC and low molecular weight solvent as a function of concentration and temperature. The results are systematic and look scientific interest, but the report is not well organized and not well described. I would like to recommend this report for publication after a major revision.

Here is my suggestions for the author:-

-        Somehow the conclusion is missing. I guess the authors merged conclusions with the discussion. I would recommend rewriting concisely and clearly.

-        I found many typos in the manuscript. Check Line No: 15, 19, 49, 58, etc.. All these typos must be corrected in the revision.

-        The microscopic technique (for instance polarizing optical microscope) or thermograms would be an efficient technique to describe the system properties at a specific phase. I suggest authors to produce these data instead of photographic images alone.

-        The writing and presentation of the entire report must be improved. I suggest authors to describe the introduction concisely and subsequent references can also be reduced.

Overall, I would recommend this paper for further consideration. Anyway, a major revision must be done before being considered for publication.

Author Response

Reviewer # 3

The Reviewer supported the paper, indicating some issues with the supplementation.

 Reviewer # 3:  ‘Somehow the conclusion is missing. I guess the authors merged conclusions with the discussion. I would recommend rewriting concisely and clearly.’

The answer:  In the revised manuscript, Conclusions section has been explicitly introduced

Reviewer # 3:  - I found many typos in the manuscript. Check Line No: 15, 19, 49, 58, etc.. All these typos must be corrected in the revision.’

The answer:  The types have been corrected

Reviewer # 3:  ‘-  The microscopic technique (for instance, polarizing optical microscope) or thermograms would be an efficient technique to describe the system properties at a specific phase. I suggest authors produce these data instead of photographic images alone.’

The answer:  The basic thermogram offers mainly subsequent phase transition temperatures, which are available from the visual detection, where also some conclusions above the nature of subsequent phases are possible. In our opinion, it is rather the task for further studies, particularly for ‘dense & in-deep’ tests offering data for the analysis of pretransitional effects. Notwithstanding, such analysis is associated with fitting via functions containing 5 – 6 ‘free’ parameters. We used NDE, directly coupled to fluctuations, where the analysis can be reduced simply to the reciprocal of experimental data.

Regarding the ‘polarizing’ insight, we introduced such pictures focusing on the two-coexisting-phases domain in the low concentrations  (see new Figure  xx and the coupled text).  The grand advantage of ‘ampoule’ observations are issues regarding critical opalescence. Namely:

  • completely unexpected critical opalescence in ‘low x’ domain
  • surprisingly large range of the critical opalescence in ‘high x’ domain
  • The ‘bluish’ part of the opalescence.

We stress all these in Conclusions.

Reviewer # 3:  ‘-  The writing and presentation of the entire report must be improved. I suggest authors to describe the introduction concisely and subsequent references can also be reduced.

The answer:  The Conclusion Sections, Sub-Sections, and general clarity test follow the Reviewer's suggestion. The reduction of references seems to be not possible. The number of references results from the fact that the paper addresses a few issues, not too often considered in one report. They are issues related to the phase diagram, pretransitional effects, description of coexistence curves, critical opalescence, and nonlinear dielectric effect.

In fact, the number of references had to be increased by 18 positions, as suggested by Reviewer#1 in the opinion (see above).

In summary: 

  • Only changes that explicitly followed (all) reviewers' comment has been introduced
  • The paper has been ‘deeply cleaned.’
  • Following comments, the new Figure (current Fig. 4) has been added.
  • Following comments ‘Results’ section has been re-organized into 4 subsection
  • The “Conclusions’ section has been added. It particularly addresses such issues as possible future studies and innovative applications – as suggested by the Reviewer .
  • There are 18 additional references, indicated by Reviewer 1.

Round 2

Reviewer 3 Report

The manuscript can be improved further.